# Patient satisfaction in outdoor department of primary health care facilities in Rohingya refugee camps in Bangladesh: A cross-sectional study

Raisul Islam[1], Tasnova Sadneen[1], Md. Shakkor Rahman[1], Mohammad Nayeem Hasan[2], Mirza Asif Adnan[3], Mohammad Nayeem[4,5], Ahmad Zubair Mahdi[3], Abu Toha Md Rezuanul Haque Bhuiyan[6], Mohammad Delwer Hossain Hawlader[1,7]*

1 Department of Public Health, North South University, Dhaka, Bangladesh, 2 Department of Statistics, Shahjalal University of Science and Technology, Sylhet, Bangladesh, 3 HMBD Foundation, Cox's Bazar, Bangladesh, 4 Partners in Health and Development (PHD), Cox's Bazar, Bangladesh, 5 Department of Pharmaceutical Sciences, North South University, Dhaka, Bangladesh, 6 Office of The Refugee Relief and Repatriation Commissioner, Cox's Bazar, Bangladesh, 7 NSU Global Health Institute (NGHI), North South University, Dhaka, Bangladesh

* mohammad.hawlader@northsouth.edu

## Abstract

### Background

Perception of patients on healthcare services is important in evaluating the quality of a healthcare structure. Focusing on a major humanitarian emergency in South-East Asia, this research aims to explore patient satisfaction regarding primary health care in Rohingya refugee camps placed in Bangladesh.

### Methods

We conducted this cross-sectional study during November 2023 – March 2024 in five randomly selected primary health care centers in five different Rohingya refugee camps of Bangladesh. Upon selection by systematic random sampling method, 810 outdoor patients were interviewed in-person by trained data collectors. We utilized a structured questionnaire based on the Patient Satisfaction Questionnaire Short Form (PSQ-18) of Marshall and Hays. We analyzed completed response of 723 patients by IBM SPSS (v. 27) checking normality and internal consistency. Mean, standard deviation and percentage were generated for all dimensions of patient satisfaction including all PSQ-18 items. Finally, we used Kruskal–Wallis test and multivariate linear regression to explore differences and associations between patient characteristics and different dimensions of patient satisfaction.

### Results

With major responses from 18–39 years (68.7%), female (81.1%) and Rohingya ethnicity (96.8%), overall satisfaction was found among nearly 90% patients. Domain

**Data availability statement:** All relevant data are within the manuscript and its Supporting information files.

**Funding:** The author(s) received no specific funding for this work.

**Competing interests:** The authors have declared that no competing interests exist.

of interpersonal manner received highest satisfaction (95.93%) whereas the lowest was observed in accessibility and convenience (84.50%) with minor variability across different satisfaction dimensions. Ethnicity, key source of household income, type of visit and perception of type to illness were found as significant predictors in most satisfaction domains ($p < 0.05$).

## Conclusions

We concluded this study with high satisfaction in outpatient care of Rohingya camps. Therefore, focused interventions should be adopted by policymakers for maintaining the high content in patients with additional attention in domains of less patient satisfaction. Extended and periodic evaluation of patient satisfaction should be conducted for upgrading the health system of Rohingya humanitarian emergency in a patient-centric approach.

## Introduction

Measuring satisfaction level of patients is essential to evaluate the structure, operations and quality of health sector [1]. The degree of patient satisfaction is widely considered as a significant indicator of the perceived quality of care, which reflects effectiveness of health services meeting expectations of beneficiaries. An increase in satisfaction of patients makes them more efficient and adherent to treatment with reduced possibility of dropout [2]. In consideration with subjectivity, factors including result of treatment, affection of nurses and doctors, standard of accommodation and diet, time required to wait for investigations, procedures for admission and discharge affect the scale of patient satisfaction to a great extent [3,4]. Interpersonal care capacity of health workers was found as the key determinant of patient satisfaction alongside special attention on personal characteristics with probable confounding impacts [5]. Patient-centric behavioral skill of physicians profoundly impacts satisfaction level of patients more than even social determinants, economic variables and patient engagement [6]. Comparatively higher levels of satisfaction were reported among Iraqi patients from young, rural, married, unemployed, low education and good health backgrounds [7].

Multiple studies conducted in South Asia explored factors influencing satisfaction levels on different provisions of health services. Rural Indian patients receiving maternal and child health services expressed high level of satisfaction, with 63.0% to 73.9% clients being satisfied with doctors alongside 73.3% satisfaction for nurses [8]. Nearly the same level of overall satisfaction (73.1%) was found among urban healthcare beneficiaries of India with concerns expressed regarding amount of time spent with physicians [9]. Satisfaction of outdoor diabetic patients in Pakistan was highly associated with their understandings on technical skills, interpersonal aspects and appropriateness of time provision [10]. In Nepal, the study patients expressed 38.9%, 45.1% and 92.1% satisfaction respectively on dimension regarding general satisfaction, accessibility and convenience, and interpersonal manner [11].

Discipline in the service environment and assurance of hospital employees in Bangladesh carried more importance than responsiveness and communication [12]. In terms of services and factors for choosing hospitals, private health facilities in Bangladesh were evaluated more favorably than public hospitals regarding responsiveness, communication and discipline [13]. Higher satisfaction was expressed in government hospitals of Bangladesh among patients being young, facing shorter waiting time, and having less education [14]. Recent study in Bangladesh highlighted gap between advice of healthcare staff and perceptions of patients, focusing on transparent communication and patient-centered medical sessions to align patients' awareness with diagnostics and treatment plans offered to them [6].

Bangladesh is a country densely populated with around 170 million host community people having 1,119 persons per square kilometer [15]. In addition, this country is providing shelter to 989,585 Rohingya refugees (an ethnic group who fled from Myanmar following series of violent incidents) across 33 densely populated camps in Cox's Bazar district alongside Bhasan Char Island with assistance of humanitarian and development partners [16]. As reported in July 2024, a total of 56 health sector partners are providing essential health services to these refugees through a network of 104 primary health care facilities [17]. Although, there is limited availability of published articles on satisfaction of patients receiving healthcare services in Rohingya camps of Bangladesh. A 2019 study conducted in Rohingya camps found the majority of respondents had good responses both for treatment and services received by them. Satisfaction was expressed by the study participants regarding behaviour, examination, explanation and advice. In contrast, dissatisfaction was reported regarding etiquette of some health staff alongside long waiting period [18]. The Public Health Needs Assessment (2024−25) reported that the health system for Rohingya refugees is burdened with key accessibility related challenges including longer waiting time, deficit of available services, limited transportation facility and linguistic barrier, all of which can affect the patient satisfaction to a great extent [19]. This public health finding emphasizes the need for in-depth analysis of patient-centered outcomes such as satisfaction for guiding quality improvement in Rohingya healthcare. Taking account of the magnitude of the Rohingya refugee crisis and high dependance on international assistance, evaluation of patient satisfaction will provide us with a snapshot of perceived quality of healthcare being provided in this resource-constrained humanitarian emergency. In our study, we aim at measuring level of satisfaction across different dimensions among patients receiving outpatient care from primary health care facilities placed in Rohingya camps of Bangladesh.

## Methodology

### Study settings and subjects

The sample size was calculated using Cochran's formula ($n = Z^2 pq/d^2$) representing Z (2.17) as the normal standard deviation corresponding to the desired 97% confidence interval, p as the estimated proportion of satisfied patients (50%), q=(1-p), with a margin of error (d) of 4%. As the true proportion is unknown for Rohingya refugee settings from earlier literatures, we presumed variability to the maximum extent; proportion of satisfied patients is considered as 50%. Estimating additional 10% for refusals or incomplete responses, sample size of 810 patients was targeted to be interviewed during the study period. We selected 97% confidence interval for achieving higher precision than the traditional 95%, remaining feasible within our technical and financial resources. Although a 99% confidence interval would generate even greater precision, we would need a substantially larger sample size, which would not be manageable considering the resource constraints of our study.

This cross-sectional study was conducted from 15 November 2023–15 March 2024 utilizing the two-stage sampling technique. Firstly, five primary health care facilities were selected randomly as study centers situated across five different Rohingya camps (2 West, 8 West, 10, 13 and 16) of Bangladesh. Within health sector response, all selected health facilities were operationalized by national non-government organizations with funding support from international donors and United Nations agencies. Secondly, systematic random sampling technique was utilized for selecting the study participants. Using a random number table, a single-digit number (3) was selected to determine the starting point. Accordingly, the third patient completing the outpatient consultation was approached as the first

study respondent. So, every third patient till the end of working hours were interviewed consecutively after obtaining written informed consent. Recruitment of human participants for this study was conducted during 1 January – 29 February 2024. We used several criteria for eligibility in selecting the study respondents which included: (a) adult patients (aged ≥18 years) completed outdoor consultations; (b) willing to participate and give consent; (c) under-stand the study questionnaire. Patients with emergency health conditions and volunteers/ staff of the selected health facilities were excluded from this study to avoid conflict of interest. The study protocol is provided as supporting information (See S2 File).

## Data collection

As we approached 810 patients as per eligibility criteria, complete information was provided by 723 participants with 89% response rate. The 87 non-respondents included participants who refused or could not complete the interview. Face to face interview technique was used to collect the required information utilizing a structured questionnaire (See S3 File). It was comprised of four segments. Starting with basic information (name and address) in the first segment, data on sociodemographic characteristics including age, sex, ethnicity, religion, marital status, education level and employment status of patients were collected using the second section of the questionnaire. Based on a survey done among Rohingya refugees [20], educational level was categorized in three categories; no literacy for participants who cannot read or write, basic literacy with reading capacity only, and functional literacy with both reading and writing skills. The next segment gathered data on health status of patients including types of facility visit and perception of category of illness. In the final section, a globally recognized survey instrument having acceptable internal consistency and reliability, the Patient Satisfaction Questionnaire (PSQ-18) was utilized to measure satisfaction level of patients [21]. A total of 18 question items were utilized in this questionnaire (See Table A in S1 File). This tool evaluated patient satisfaction in seven sub-scales including general satisfaction, technical quality, interpersonal manner, communication, financial aspects, time spent with the doctor, accessibility and convenience. In addition, overall patient satisfaction was also measured accumulating responses from all seven sub-scales. Linguistic experts in our research team, with prior experience of working with the PSQ-18 questionnaire, were involved in the Bengali translation of our study questionnaire. The final version was developed through consensus among the research team ensuring appropriate contextualization. A field team of five Bangladeshi non-health personnel were deployed as enumerators. For facilitating comfortable and engaging communication with study participants, we selected data collectors with substantial work experience in Rohingya camps, who were fluent in the local dialect of Chattogram, which carries substantial speaking similarity with the language used in Rohingya camps. In addition, Rohingya community leaders related to study sites, known as 'Majhi', were informed about the study background, objectives, data collection process including voluntary nature of participation, data confidentiality and expected outcomes. Formal training was provided to enumerators on structures of the questionnaire, interview techniques and ethical issues. In each of study sites, each enumerator approached outdoor patients to conduct approximately 20 interviews per day following the methodology. To avoid conflict of professional and organizational interest, any employee of study health facilities was not considered for the role of enumerator.

## Data analysis

Following collection of all responses form paper-based questionnaires, we entered and analyzed data in IBM Statistical Package for the Social Sciences (SPSS) version 27. For all question items of PSQ-18, responses were structured across five types; 'Strongly Agree', 'Agree', 'Uncertain', 'Disagree' and 'Strongly Disagree' with values ranging from 1 to 5 (See Table B in S1 File). Question items which are worded negatively, the scale value was reversed following scoring guidelines of PSQ-18. Completing scores of all items, average was done together for items within same domain to generate scores for the targeted domains (See Table C in S1 File).

Independent variables including sociodemographic and health information of respondents were analyzed in frequencies and percentages. Tests for normality of data were conducted for dependent variables using Kolmogorov-Smirnov test and Shapiro-Wilk test. To check internal consistency of survey questionnaire, reliability tests were utilized to measure Cronbach's alpha. Descriptive statistics was used for generating mean, standard deviation and percentage for all dimensions of patient satisfaction including items of PSQ-18 questionnaire. Mean differences in patient satisfaction across subgroups of independent variables were explored through nonparametric Kruskal–Wallis test with significance level 0.05 and comparison of means. Finally, multivariate linear regression was conducted for identifying independent predictors of patient satisfaction across different dimensions. The de-identified dataset used for analysis is available as supporting information (See S4 File).

## Ethical approval

We obtained ethical approval for this study from Institutional Review Board/ Ethics Review Committee (IRB/ERC), North South University, Bangladesh with the reference number 2023/OR-NSU/IRB/1238. In addition, administrative approval was obtained from Office of The Refugee Relief and Repatriation Commissioner, Cox's Bazar. Before undertaking interviews, written informed consent was obtained from each willing respondent ensuring respect for right to informed consent, right to deny, and right to accurate representation. Although the consent form was in English language, it was read out and explained in detail to study participants in local dialect. Voluntary nature of participation was informed to every respondent and option was given to withdraw from investigation at any time, without justification, and without any consequence. All collected information were handled with utmost confidentiality and use was limited to only research purpose.

## Results

### Socio-demographic and health characteristics

Sociodemographic and health information of the respondents are displayed in Table 1. Mean age of the participants was found 35.7 (SD ± 15.0) with major representation from 18–39 years category (68.7%). Females constituted more than four-fifth of study participants (81.1%). Over ninety percent of respondents were from Rohingya community (96.8%) and Islam religion (99.6%).

Greater than three quarter participants were reported as married (85.5%), with no literacy level (82.6%) who could not read neither write. Regarding key source of household income, nearly one third of respondents depended solely on humanitarian assistance (32.4%) which was followed by daily labour (29.0%), NGO (non-government organization) volunteer (14.2%) and small business (11.2%) as notable proportion. Most of the patients visited the health centers for follow-up consultation (64.2%) with acute or sudden onset illness (62.9%) in terms of their perception.

### Reliability and patient satisfaction across different dimensions

Table 2 represents data on reliability of subscales of PSQ-18 tool alongside satisfaction of patients across different domains. With an overall internal consistency of 0.79, Cronbach's alpha for all seven subscales was found within range of 0.6 and 0.8, making the tool moderately acceptable [22]. Among the dimensions of patient satisfaction, out of maximum mean score 5, interpersonal manner came up with highest standing (4.80 ± 0.40) with 95.93% satisfied patients. On the other hand, least satisfaction was expressed in the domain of accessibility and convenience (4.23 ± 0.72) with satisfaction level of 84.50%. Mean score for the overall patient satisfaction was found (4.52 ± 0.36) and 90.37% study participants were found satisfied in this regard.

### Satisfaction of respondents across all items of PSQ-18

Information related to patient satisfaction across all items of PSQ-18 are presented in Table 3. Regarding the dimension of general satisfaction, almost 92% of respondents experienced near perfection about the healthcare they received

**Table 1. Sociodemographic and health information of respondents (n = 723).**

| Characteristic | Category | Frequency | Percentage |
|---|---|---|---|
| Mean Age in Years (SD) | – | 35.7 (15.0) | – |
| Age (Years) | 18-39 | 497 | 68.7% |
| | 40-59 | 140 | 19.4% |
| | ≥60 | 86 | 11.9% |
| Sex | Female | 586 | 81.1% |
| | Male | 137 | 18.9% |
| Ethnicity | Rohingya | 700 | 96.8% |
| | Bangladeshi | 23 | 3.2% |
| Religion | Islam | 720 | 99.6% |
| | Others | 3 | 0.4% |
| Marital Status | Married | 618 | 85.5% |
| | Widowed | 59 | 8.2% |
| | Unmarried | 35 | 4.8% |
| | Divorced | 11 | 1.5% |
| Educational Level | No Literacy | 597 | 82.6% |
| | Basic Literacy | 64 | 8.9% |
| | Functional Literacy | 62 | 8.6% |
| Key Source of Household Income | Daily Labour | 210 | 29.0% |
| | NGO Volunteer | 103 | 14.2% |
| | Small Business | 81 | 11.2% |
| | Teacher | 33 | 4.6% |
| | Masonry | 31 | 4.3% |
| | Tailor | 5 | 0.7% |
| | Others | 26 | 3.6% |
| | Depends only on humanitarian aids | 234 | 32.4% |
| Type of Visit | New Visit | 259 | 35.8% |
| | Follow-up Visit | 464 | 64.2% |
| Perception of type of Illness | Acute | 455 | 62.9% |
| | Chronic | 268 | 37.1% |

**Table 2. Reliability of subscales of PSQ-18 tool with satisfaction of patients across different domains (n = 723).**

| Domains of Patient Satisfaction | Cronbach's alpha | Maximum Mean | Mean Score | Standard Deviation | Satisfaction (%) |
|---|---|---|---|---|---|
| General Satisfaction | 0.61 | 5 | 4.48 | 0.67 | 89.52% |
| Technical Quality | 0.61 | 5 | 4.46 | 0.48 | 89.13% |
| Interpersonal Manner | 0.69 | 5 | 4.80 | 0.40 | 95.93% |
| Communication | 0.64 | 5 | 4.70 | 0.49 | 94.05% |
| Financial Aspects | 0.78 | 5 | 4.72 | 0.69 | 94.50% |
| Time Spent with Doctor | 0.65 | 5 | 4.60 | 0.53 | 92.03% |
| Accessibility and Convenience | 0.61 | 5 | 4.23 | 0.72 | 84.50% |
| Overall Patient Satisfaction | 0.79 | 5 | 4.52 | 0.36 | 90.37% |

**Table 3. Satisfaction of study participants across all items of PSQ-18 (n = 723).**

| Item No. | Questions of PSQ-18 | Agreement (Strongly Agree + Agree) Frequency (%) | Uncertain Frequency (%) | Disagreement (Strongly Disagree + Disagree) Frequency (%) | Mean Score | Standard Deviation |
|---|---|---|---|---|---|---|
| 1 | Doctors are good about explaining the reason for medical tests | 715 (98.89%) | 1 (0.14%) | 7 (0.97%) | 4.65 | 0.55 |
| 2 | I think my doctor's office has everything needed to provide complete medical care | 583 (80.64%) | 101 (13.97%) | 39 (5.39%) | 4.08 | 0.90 |
| 3 | The medical care I have been receiving is just about perfect | 663 (91.70%) | 55 (7.61%) | 5 (0.69%) | 4.43 | 0.67 |
| 4 | Sometimes doctors make me wonder if their diagnosis is correct | 32 (4.43%) | 9 (1.24%) | 682 (94.33%) | 4.47 | 0.74 |
| 5 | I feel confident that I can get the medical care I need without being set back financially | 689 (95.30%) | 8 (1.11%) | 26 (3.60%) | 4.69 | 0.80 |
| 6 | When I go for medical care, they are careful to check everything when treating and examining me | 718 (99.31%) | 1 (0.14%) | 4 (0.55%) | 4.62 | 0.53 |
| 7 | I have to pay for more of my medical care than I can afford | 31 (4.29%) | 0 (0.00%) | 692 (95.71%) | 4.76 | 0.72 |
| 8 | I have easy access to the medical specialists I need | 562 (77.73%) | 24 (3.32%) | 137 (18.95%) | 3.87 | 1.38 |
| 9 | Where I get medical care, people have to wait too long for emergency treatment | 141 (19.50%) | 60 (8.30%) | 522 (72.20%) | 4.04 | 1.19 |
| 10 | Doctors act too businesslike and impersonal toward me | 7 (0.97%) | 0 (0.00%) | 716 (99.03%) | 4.87 | 0.42 |
| 11 | My doctors treat me in a very friendly and courteous manner | 718 (99.31%) | 0 (0.00%) | 5 (0.69%) | 4.72 | 0.50 |
| 12 | Those who provide my medical care sometimes hurry too much when they treat me | 19 (2.63%) | 4 (0.55%) | 700 (96.82%) | 4.66 | 0.63 |
| 13 | Doctors sometimes ignore what I tell them | 18 (2.49%) | 6 (0.83%) | 699 (96.68%) | 4.75 | 0.59 |
| 14 | I have some doubts about the ability of the doctors who treat me | 8 (1.11%) | 19 (2.63%) | 696 (96.27%) | 4.65 | 0.60 |
| 15 | Doctors usually spend plenty of time with me | 709 (98.06%) | 2 (0.28%) | 12 (1.66%) | 4.54 | 0.61 |
| 16 | I find it hard to get an appointment for medical care right away | 77 (10.65%) | 2 (0.28%) | 644 (89.07%) | 4.34 | 0.95 |
| 17 | I am dissatisfied with some things about the medical care I receive | 59 (8.16%) | 2 (0.28%) | 662 (91.56%) | 4.52 | 0.90 |
| 18 | I am able to get medical care whenever I need it | 717 (99.17%) | 1 (0.14%) | 5 (0.69%) | 4.64 | 0.52 |

whereas just over 8% disagreed in this case. In the aspect of technical quality, a significant proportion of study participants were found content regarding capacity of physician's office (80.64%) and care during treatment (99.31%). On the contrary, only a handful of patients were doubtful on diagnosis (4.43%) and technical capability (1.11%) of doctors. For interpersonal manner, almost all respondents appreciated manner of the physicians in terms of friendliness and courtesy (99.31%).

In relation to communication, most of the patients praised doctors in elaborating the rationale for clinical investigations (98.89%) whereas fewer than 3% participants agreed on the accusation that physicians sometimes remain ignorant on what patients inform them. In terms of financial capacity, nearly all participants felt confident to get the medical care without financial setback (95.30%) while less than 5% patients had to expense for healthcare more than they can afford. Regarding domain of time spent with doctor, nearly 98% respondents agreed that physicians often attend them with a lot

of time. For accessibility and convenience, majority of study participants easily accessed to the required medical specialists (77.73%) whereas around 11% respondents struggled to secure an appointment for healthcare immediately.

## Differences in patient satisfaction across sociodemographic characteristics and health-related subgroups

Table 4 summarizes differences in level of patient contentment across sociodemographic characteristics and health-related subgroups. We found no statistically significant difference in patient satisfaction across sex, religion and marital status in any of the core dimensions in this study. General satisfaction of patients significantly differed among age groups ($p = 0.003$) with highest score for respondents aged 18–39 years ($4.53 \pm 0.64$) in comparison with other age categories. Except for technical quality and financial aspects, ethnicity demonstrated statistically significant differences in level of satisfaction across all other domains of patient satisfaction with P-value < 0.05. Study participants with Rohingya ethnicity were found less satisfied comparing to Bangladeshi nationals regarding general satisfaction ($4.47 \pm 0.68$), interpersonal manner ($4.79 \pm 0.41$), communication ($4.69 \pm 0.49$), time spent with doctor ($4.59 \pm 0.54$), accessibility and convenience ($4.21 \pm 0.72$) alongside overall satisfaction ($4.51 \pm 0.36$).

Across different levels of education, statistically significant differences in satisfaction were found regarding technical quality ($p = 0.036$), interpersonal manner ($p = 0.031$), financial aspects ($p = 0.002$), time spent with doctor ($p = 0.005$), accessibility and convenience ($p < 0.001$) beside overall satisfaction ($p < 0.001$). Respondents with functional literacy (who can read and write) expressed higher level of patient satisfaction across all dimensions. Statistically, key source of household income showed significant differences in all dimensions of patient satisfaction except for accessibility and convenience ($p = 0.148$). Respondents depending only on humanitarian assistance, were less content regarding general satisfaction and ($4.26 \pm 0.72$) and technical quality ($4.35 \pm 0.49$) in comparison with occupational sources of household income. Regarding type of visits to health centers, statistically significant difference was found in all dimensions of patient satisfaction ($p < 0.05$) where follow-up consultations yielded better satisfaction in contrast to new visits in every domain. For perception of type of illness, level of patient satisfaction did not vary significantly in terms of communication and financial aspects. Respondents visiting health centers with perception of chronic/long-term illness was found less satisfied comparing to patients with acute/sudden-onset health issues except for accessibility and convenience ($4.31 \pm 0.70$).

## Association between sociodemographic characteristics and health information with patient satisfaction domains (n=723)

In Table 5, multivariate linear regression was performed for exploring the association of key sociodemographic and health-related characteristics with different dimensions of patient satisfaction. We observed ethnicity having significantly positive association with most of the domains of patient satisfaction, including general satisfaction (Coef. = 0.10, p = 0.007), technical quality (Coef. = 0.12, p < 0.001), interpersonal manner (Coef. = 0.11, p = 0.004), communication (Coef. = 0.10, p = 0.009), time spent with doctor (Coef. = 0.14, p < 0.001), accessibility and convenience (Coef. = 0.14, p < 0.001), and overall satisfaction (Coef. = 0.18, p < 0.001), with the highest impact reflected in the overall satisfaction domain. Type of visit was found as the most consistent predictor, with a positive association with all dimensions of patient satisfaction, having major influence on overall satisfaction (Coef. = 0.35, p < 0.001) and technical quality (Coef. = 0.31, p < 0.001). In addition, key source of household income appeared with strong positive association, notably regarding overall satisfaction (Coef. = 0.16, p < 0.001) and interpersonal manner (Coef. = 0.18, p < 0.001).

Conversely, perception of type of illness was found negatively associated with patient satisfaction, having significant impact on general satisfaction (Coef. = −0.26, p < 0.001) and overall satisfaction (Coef. = −0.20, p < 0.001). Negative association was also observed in case of marital status with general satisfaction (Coef. = −0.10, p = 0.009), whereas age was positively associated only with time spent with doctor (Coef. = 0.11, p = 0.011). We found no statistically significant association for sex, religion, or educational level in any dimension of patient satisfaction.

 

Table 4. Differences in patient satisfaction across sociodemographic characteristics and health-related subgroups (n = 723).

| Characteristic | Category | General Satisfaction | | Technical Quality | | Interpersonal Manner | | Communication | | Financial Aspects | | Time Spent with Doctor | | Accessibility and Convenience | | Overall Satisfaction | |
|---|---|---|---|---|---|---|---|---|---|---|---|---|---|---|---|---|---|
| | | P-Value | Mean (SD) | P-Value | Mean (SD) | P-Value | Mean (SD) | P-Value | Mean (SD) | P-Value | Mean (SD) | P-Value | Mean (SD) | P-Value | Mean (SD) | P-Value | Mean (SD) |
| **Age in Years** | 18-39 | 0.003 | 4.53 (0.64) | 0.343 | 4.47 (0.47) | 0.153 | 4.82 (0.36) | 0.424 | 4.71 (0.48) | 0.614 | 4.73 (0.65) | 0.465 | 4.59 (0.54) | 0.072 | 4.24 (0.71) | 0.2 | 4.53 (0.36) |
| | 40-59 | | 4.37 (0.76) | | 4.39 (0.52) | | 4.76 (0.53) | | 4.66 (0.56) | | 4.73 (0.71) | | 4.61 (0.54) | | 4.12 (0.77) | | 4.46 (0.39) |
| | ≥60 | | 4.36 (0.66) | | 4.48 (0.44) | | 4.75 (0.38) | | 4.76 (0.43) | | 4.67 (0.87) | | 4.67 (0.44) | | 4.33 (0.69) | | 4.54 (0.34) |
| **Sex** | Female | 0.123 | 4.49 (0.66) | 0.091 | 4.47 (0.48) | 0.281 | 4.81 (0.38) | 0.835 | 4.70 (0.50) | 0.404 | 4.73 (0.66) | 0.645 | 4.60 (0.54) | 0.386 | 4.23 (0.73) | 0.232 | 4.52 (0.37) |
| | Male | | 4.42 (0.70) | | 4.41 (0.46) | | 4.76 (0.49) | | 4.72 (0.45) | | 4.71 (0.81) | | 4.61 (0.48) | | 4.21 (0.68) | | 4.49 (0.34) |
| **Ethnicity** | Rohingya | 0.013 | 4.47 (0.68) | 0.075 | 4.45 (0.48) | 0.002 | 4.79 (0.41) | 0.001 | 4.69 (0.49) | 0.698 | 4.73 (0.69) | < 0.001 | 4.59 (0.54) | < 0.001 | 4.21 (0.72) | < 0.001 | 4.51 (0.36) |
| | Bangladeshi | | 4.76 (0.47) | | 4.64 (0.33) | | 5.00 (0.00) | | 4.98 (0.10) | | 4.72 (0.78) | | 4.96 (0.14) | | 4.68 (0.40) | | 4.79 (0.19) |
| **Religion** | Islam | 0.612 | 4.48 (0.67) | 0.459 | 4.46 (0.48) | 0.953 | 4.80 (0.40) | 0.191 | 4.70 (0.49) | 0.316 | 4.72 (0.69) | 0.814 | 4.60 (0.53) | 0.077 | 4.22 (0.72) | 0.288 | 4.52 (0.36) |
| | Others | | 4.67 (0.58) | | 4.33 (0.14) | | 4.83 (0.29) | | 5.00 (0.00) | | 5.00 (0.00) | | 4.67 (0.58) | | 4.83 (0.29) | | 4.72 (0.24) |
| **Marital Status** | Married | 0.097 | 4.50 (0.64) | 0.176 | 4.46 (0.47) | 0.697 | 4.80 (0.42) | 0.737 | 4.70 (0.49) | 0.061 | 4.72 (0.69) | 0.518 | 4.60 (0.54) | 0.413 | 4.21 (0.72) | 0.423 | 4.52 (0.36) |
| | Widowed | | 4.43 (0.74) | | 4.51 (0.49) | | 4.81 (0.31) | | 4.65 (0.60) | | 4.81 (0.63) | | 4.65 (0.46) | | 4.31 (0.73) | | 4.55 (0.39) |
| | Unmarried | | 4.27 (0.89) | | 4.38 (0.43) | | 4.79 (0.28) | | 4.79 (0.33) | | 4.93 (0.18) | | 4.51 (0.52) | | 4.35 (0.56) | | 4.53 (0.32) |
| | Divorced | | 3.91 (1.07) | | 4.16 (0.73) | | 4.82 (0.25) | | 4.68 (0.34) | | 4.18 (1.59) | | 4.73 (0.41) | | 3.98 (0.90) | | 4.29 (0.56) |
| **Educational Level** | No Literacy | 0.194 | 4.47 (0.67) | 0.036 | 4.47 (0.48) | 0.031 | 4.79 (0.41) | 0.073 | 4.70 (0.49) | 0.002 | 4.76 (0.62) | 0.005 | 4.60 (0.53) | < 0.001 | 4.25 (0.72) | < 0.001 | 4.52 (0.36) |
| | Basic Literacy | | 4.39 (0.70) | | 4.32 (0.48) | | 4.74 (0.36) | | 4.64 (0.57) | | 4.30 (1.19) | | 4.43 (0.65) | | 3.91 (0.71) | | 4.33 (0.36) |
| | Functional Literacy | | 4.57 (0.68) | | 4.50 (0.45) | | 4.87 (0.34) | | 4.81 (0.38) | | 4.78 (0.52) | | 4.76 (0.38) | | 4.35 (0.60) | | 4.61 (0.33) |

*(Continued)*

Table 4. (Continued)

| Characteristic | Category | General Satisfaction | | Technical Quality | | Interpersonal Manner | | Communication | | Financial Aspects | | Time Spent with Doctor | | Accessibility and Convenience | | Overall Satisfaction | |
|---|---|---|---|---|---|---|---|---|---|---|---|---|---|---|---|---|---|
| | | P-Value | Mean (SD) | P-Value | Mean (SD) | P-Value | Mean (SD) | P-Value | Mean (SD) | P-Value | Mean (SD) | P-Value | Mean (SD) | P-Value | Mean (SD) | P-Value | Mean (SD) |
| **Key Source of Household Income** | | < 0.001 | | < 0.001 | | < 0.001 | | < 0.001 | | 0.037 | | 0.001 | | 0.148 | | < 0.001 | |
| | Daily Labour | | 4.51 (0.65) | | 4.49 (0.46) | | 4.76 (0.49) | | 4.65 (0.55) | | 4.85 (0.39) | | 4.57 (0.54) | | 4.26 (0.67) | | 4.54 (0.34) |
| | NGO Volunteer | | 4.64 (0.63) | | 4.51 (0.50) | | 4.89 (0.24) | | 4.81 (0.38) | | 4.69 (0.67) | | 4.62 (0.53) | | 4.29 (0.71) | | 4.58 (0.37) |
| | Small Business | | 4.54 (0.57) | | 4.48 (0.48) | | 4.85 (0.29) | | 4.72 (0.45) | | 4.71 (0.69) | | 4.65 (0.56) | | 4.30 (0.67) | | 4.56 (0.35) |
| | Teacher | | 4.71 (0.31) | | 4.54 (0.38) | | 4.95 (0.19) | | 4.88 (0.25) | | 4.50 (1.09) | | 4.80 (0.43) | | 4.26 (0.71) | | 4.60 (0.30) |
| | Masonry | | 4.60 (0.89) | | 4.53 (0.54) | | 4.98 (0.09) | | 4.90 (0.20) | | 4.40 (1.16) | | 4.71 (0.56) | | 4.13 (0.94) | | 4.55 (0.49) |
| | Tailor | | 4.60 (0.42) | | 4.60 (0.52) | | 4.70 (0.45) | | 4.60 (0.42) | | 4.40 (0.42) | | 4.30 (0.84) | | 3.70 (0.87) | | 4.36 (0.12) |
| | Others | | 4.83 (0.28) | | 4.69 (0.32) | | 4.94 (0.16) | | 4.87 (0.27) | | 4.90 (0.20) | | 4.83 (0.34) | | 4.47 (0.51) | | 4.74 (0.21) |
| | Depends only on humanitarian aids | | 4.26 (0.72) | | 4.35 (0.49) | | 4.71 (0.43) | | 4.63 (0.53) | | 4.70 (0.78) | | 4.54 (0.52) | | 4.13 (0.76) | | 4.42 (0.36) |
| **Type of Visit** | | < 0.001 | | < 0.001 | | 0.001 | | < 0.001 | | < 0.001 | | 0.006 | | < 0.001 | | < 0.001 | |
| | New Visit | | 4.39 (0.65) | | 4.31 (0.44) | | 4.75 (0.39) | | 4.65 (0.47) | | 4.53 (0.94) | | 4.56 (0.49) | | 3.98 (0.74) | | 4.38 (0.33) |
| | Follow-up Visit | | 4.53 (0.68) | | 4.54 (0.48) | | 4.82 (0.41) | | 4.73 (0.50) | | 4.83 (0.47) | | 4.63 (0.55) | | 4.36 (0.67) | | 4.59 (0.36) |
| **Perception of type of Illness** | | < 0.001 | | < 0.001 | | < 0.001 | | 0.152 | | 0.356 | | 0.03 | | 0.007 | | 0.024 | |
| | Acute | | 4.59 (0.56) | | 4.51 (0.44) | | 4.84 (0.37) | | 4.72 (0.48) | | 4.75 (0.56) | | 4.63 (0.52) | | 4.17 (0.73) | | 4.54 (0.34) |
| | Chronic | | 4.28 (0.80) | | 4.36 (0.53) | | 4.73 (0.44) | | 4.68 (0.50) | | 4.67 (0.87) | | 4.55 (0.55) | | 4.31 (0.70) | | 4.47 (0.40) |

**Table 5. Association between sociodemographic characteristics and health information with patient satisfaction domains (n = 723).**

| Characteristic | General Satisfaction | | Technical Quality | | Interpersonal Manner | | Communication | | Financial Aspects | | Time Spent with Doctor | | Accessi-bility and Convenience | | Overall Satisfaction | |
|---|---|---|---|---|---|---|---|---|---|---|---|---|---|---|---|---|
| | Coef. | P-Value | Coef. | P-Value | Coef. | P-Value | Coef. | P-Value | Coef. | P-Value | Coef. | P-Value | Coef. | P-Value | Coef. | P-Value |
| Age in Years | −0.01 | 0.787 | 0.06 | 0.131 | −0.01 | 0.974 | 0.04 | 0.311 | −0.03 | 0.425 | 0.107 | 0.011 | −0.01 | 0.875 | 0.03 | 0.455 |
| Sex | −0.01 | 0.734 | −0.04 | 0.37 | −0.03 | 0.434 | 0.01 | 0.89 | 0.05 | 0.261 | −0.03 | 0.437 | 0.02 | 0.682 | −0.04 | 0.911 |
| Ethnicity | 0.1 | 0.007 | 0.12 | < 0.001 | 0.11 | 0.004 | 0.1 | 0.009 | 0.04 | 0.287 | 0.14 | < 0.001 | 0.14 | < 0.001 | 0.18 | < 0.001 |
| Religion | 0.03 | 0.94 | −0.03 | 0.407 | −0.02 | 0.562 | 0.02 | 0.679 | 0.05 | 0.224 | −0.03 | 0.505 | 0.05 | 0.183 | 0.02 | 0.616 |
| Marital Status | −0.1 | 0.009 | −0.04 | 0.309 | 0.03 | 0.395 | 0.02 | 0.569 | 0.02 | 0.639 | 0.01 | 0.804 | 0.03 | 0.399 | −0.04 | 0.905 |
| Educational Level | −0.03 | 0.439 | −0.04 | 0.387 | −0.05 | 0.298 | −0.03 | 0.513 | −0.06 | 0.206 | −0.02 | 0.638 | −0.05 | 0.222 | −0.07 | 0.111 |
| Key Source of Household Income | 0.17 | < 0.001 | 0.13 | < 0.001 | 0.18 | < 0.001 | 0.17 | < 0.001 | −0.07 | 0.068 | 0.13 | < 0.001 | 0.07 | 0.078 | 0.16 | < 0.001 |
| Type of Visit | 0.18 | < 0.001 | 0.31 | < 0.001 | 0.13 | < 0.001 | 0.1 | 0.014 | 0.26 | < 0.001 | 0.1 | 0.013 | 0.26 | < 0.001 | 0.35 | < 0.001 |
| Perception of type of Illness | −0.26 | < 0.001 | −0.25 | < 0.001 | −0.15 | < 0.001 | −0.06 | 0.134 | −0.15 | < 0.001 | −0.12 | 0.004 | 0.01 | 0.759 | −0.2 | < 0.001 |

## Discussion

Overall patient satisfaction (90.37%) measured in this research was found largely higher comparing to similar studies [9,23,24]. Despite increasing operational and financial constraints in Rohingya health care system, this surprisingly high level of patient satisfaction might be partly explained by the contrast with extreme deprivation of Rohingyas to basic health needs in their home country. Moreover, the structured health service delivery of the study sites, operationalized by national non-government organizations with financial support from international agencies, might also be another major contributing factor alongside effective health sector coordination mechanism in Rohingya response. It reaffirms the credibility of humanitarian actors in shaping patient experience and can be considered as an example of quality benchmark in similar emergency settings. From our research, around 90% study participants were found content in the dimension of general satisfaction which was higher than findings from similar studies conducted in North London (71%) [25], Nepal (72.66%) [26] and Netherlands (80.9%) [27]. Satisfaction level in our research was also found higher than earlier similar study in Rohingya camps, which revealed satisfaction among 73.3% respondents including 59% good responses and 14.3% very good responses regarding treatment and services they received [18]. Adequate availability of essential health amenities in the study health facilities of Rohingya camps alongside improved practice of humanitarian principles by health agencies might have raised general satisfaction in comparison with other studies. Similarly, level of patient satisfaction regarding technical quality was 89.13% among respondents which may be due to adoption of standardized diagnosis and treatment approach by health care workers following national protocols or international guidelines in the study sites.

Dimension of interpersonal manner resulted in highest level of patient satisfaction (95.93%) in comparison to other domains of our study which was close to the findings from a previous study [11] whereas much lower satisfaction level was found by others (67–72.75%) [9,23]. Patient-centric behavioural approach of physicians working in Rohingya camps with reflection of humanitarian principles in daily manner might be the key driving factor in this aspect. This skillset may also be the notable factor for high satisfaction regarding communication (94.05%) and time spent with doctor (92.03%) whereas earlier studies revealed 73.43–78.8% satisfaction regarding communication and 59.07–69.71% satisfaction in the other domain [9,11,23]. Although, previous study in Rohingya camps found some sort of dissatisfaction among respondents regarding availability and behaviour of doctors, on top of their hastiness during patient consultation [18].

In case of financial aspect, despite gradual reduction of aids for Rohingya response, 94.5% of study participants expressed satisfaction as the formal health services including consultations, diagnostic tests and medicines are provided free of cost by health sector partners. Level of patient satisfaction regarding accessibility and convenience was found lowest (84.5%) in comparison to other dimensions. Limited availability of specialist physicians in Rohingya health settings and administrative formalities regarding approval for referral in higher health centers outside the camps might be the major contributing factors in this regard.

Women constituted large majority of outdoor patients participated in the study (81.1%). During the daytime, when outpatient consultations take place, male Rohingya beneficiaries are typically more engaged in income-generating activities and collection of relief items from humanitarian agencies, making them less available to seek outpatient health service. Higher proportion of female beneficiaries in outpatient health services was also seen in previous studies related to patient satisfaction [7,9,28,29]. Mean age of the study participants was found 35.7 which was found lower than studies carried out by others [26,30]. As some of the Rohingya camps are near to host community of Bangladesh, a minor proportion of respondents were with Bangladesh origin (3.2%) in our study.

Except for the domain of general satisfaction, patient satisfaction did not vary significantly across different age categories, unlike earlier studies [29–32] stating higher satisfaction among elderly patients. Although, only for time spent with doctor, a statistically significant positive association was found with age (Coef.=0.11, $p$=0.011) in our multivariate analysis. Across different sexes, our study showed absence of statistically significant difference in satisfaction of patients unlike other researchers [29–31]. During bivariate analysis, significant differences were observed across different levels of education in terms of patient satisfaction. Highest level of satisfaction was experienced by respondents with functional literacy which was contrary to the findings from previous studies [30,32] where patients with no literacy or basic education expressed more satisfaction. However, during multivariate linear regression in our study, no significant association was found between educational categories and level of patient satisfaction. Across categories of marital status, no statistically significant difference was found in level of patient satisfaction contrasting results from similar studies [30,32], while a negative association between marital status and general satisfaction was reported in our multivariate model (Coef.=−0.10, $p$=0.009). On the contrary, key source of household income, type of visit (new or follow-up) and perception of type of illness (acute or chronic) were revealed as consistent predictors across most of the dimensions of patient satisfaction. Among these variables, type of visit to health facility remained as the most consistent factor, having positive influence over all dimension, particularly over technical quality and overall satisfaction. This study revealed minor variation in satisfaction of patients across different dimensions which was consistent with the findings from others [9,33]; availability of a wide range of essential health services in Rohingya health settings might be the key contributing factor in this regard. Homogeneity in living condition, socioeconomic status and educational background among Rohingya refugees might have contributed to lack of significant associations for some key variables including age (except for time spent with doctor), sex, education and marital status (except for general satisfaction). Additionally, coordinated health sector response of stakeholders with standardized provision of essential health services likely reduced variability in patient satisfaction across these demographic categories.

The high response rate (89%) in our study ensured significant engagement, limiting concerns regarding non-response bias. Moreover, the large and representative sample size (n=723) across five different Rohingya refugee camps make the study findings more generalized. Use of an internationally acknowledged tool (PSQ-18) in our research brings reliable, consistent and diversified measurements from multiple dimensions of patient satisfaction. Most importantly, conducting this study in humanitarian settings has added real world experience both for researchers and policymakers on how to improve patient centered healthcare in a resource limited setting.

Although our research generated new insights in the domain of patient satisfaction on outpatient care in refugee camps of Bangladesh, there are few limitations to this study. Firstly, as this is a study with cross-sectional design, the findings do not necessarily relate the variables in a causal relationship. Secondly, no secondary field hospital was included as study

site in our research design, restricting generalizability of findings across all tiers of available health services in Rohingya camps. Thirdly, we did not assess perception of healthcare providers, provision of emergency allowance or incentives for working in humanitarian settings, and working environment inside Rohingya camps, which can indirectly shape the patient experience, as the motivation of service providers is a key factor in ensuring good quality of care. Availability of essential diagnostics and medicines for beneficiaries was also not evaluated in this study, which could have emerged as strong predictor of patient satisfaction. Moreover, we did not assess interventional variables including level of health awareness, access to health information, or engagement with national and international organizations, which could have generated valuable underlying factors influencing patient satisfaction in the context of humanitarian emergency.

Despite these limitations, findings from this study carry key practical implications for health services provided in humanitarian context. Higher level of patient satisfaction, notable across domains like interpersonal manner, communication, time spent with doctor, reaffirms the significance of keeping a respectful relationship between patients and healthcare workers, even in resource-limited health settings. Systematic analysis of patient satisfaction guides policy makers of health system in addressing gaps in perception to healthcare, innovating field-based interventions, ensuring effective allocation of resources and strengthening of community feedback mechanism.

## Conclusion

This research revealed high level of satisfaction among the beneficiaries receiving outpatient care in health facilities of Rohingya refugee camps. Highest satisfaction was found in the domain of interpersonal manner whereas least satisfaction was in the dimension of accessibility and convenience. Ethnicity, key source of household income, type of visit to health center and category of illness were found as statistically significant predictors for multiple dimensions of patient satisfaction. Findings from this study are expected to supplement the strengthening of humanitarian response aiming for improved patient centric health services in Rohingya camp settings. Periodically, similar type of research needs to be carried out across different tiers of health centers in Rohingya refugee camps for generating evidence-based insights on satisfaction of patients.

## Supporting information

**S1 File. PSQ-18 scoring system.**
(PDF)

**S2 File. Study protocol.**
(PDF)

**S3 File. Questionnaire.**
(PDF)

**S4 File. Dataset.**
(XLSX)

## Acknowledgments

We would like to express our earnest gratitude to the study participants for their patience and cooperation during the face-to-face interviews. We also acknowledge the valuable contributions of Jone Barua, Jitu Barua, Shuvo Barua, Apolu Barua and Himon Barua for their dedicated support in data collection and facilitation of field coordination during the study period. Our appreciation further extends to Partners in Health and Development (PHD) and HMBD Foundation for their support in implementing and facilitating the study.

## Author contributions

**Conceptualization:** Raisul Islam, Tasnova Sadneen, Md. Shakkor Rahman, Abu Toha Md Rezuanul Haque Bhuiyan, Mohammad Delwer Hossain Hawlader.

**Data curation:** Raisul Islam, Tasnova Sadneen.

**Formal analysis:** Raisul Islam, Mohammad Nayeem Hasan.

**Investigation:** Raisul Islam, Tasnova Sadneen.

**Methodology:** Raisul Islam, Tasnova Sadneen, Md. Shakkor Rahman, Mohammad Nayeem Hasan, Mohammad Delwer Hossain Hawlader.

**Project administration:** Raisul Islam, Mirza Asif Adnan, Mohammad Nayeem, Ahmad Zubair Mahdi.

**Supervision:** Mohammad Delwer Hossain Hawlader.

**Visualization:** Raisul Islam.

**Writing – original draft:** Raisul Islam.

**Writing – review & editing:** Raisul Islam, Tasnova Sadneen, Md. Shakkor Rahman, Mohammad Nayeem Hasan, Mirza Asif Adnan, Mohammad Nayeem, Ahmad Zubair Mahdi, Abu Toha Md Rezuanul Haque Bhuiyan, Mohammad Delwer Hossain Hawlader.

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
