## [Decision Letter · Decision Letter 0]

6 Mar 2025

Dear Dr. Hawlader,

Thank you for submitting your manuscript to PLOS ONE. After careful consideration, we feel that it has merit but does not fully meet PLOS ONE’s publication criteria as it currently stands. Therefore, we invite you to submit a revised version of the manuscript that addresses the points raised during the review process.

Please address each of the reviewers' comments, with particular emphasis on clarifying the methodology, as I found certain aspects ambiguous.

We look forward to receiving your revised manuscript.

Kind regards,

Md. Nuruzzaman Khan

Academic Editor

PLOS ONE

Reviewers' comments:

Reviewer's Responses to Questions

**Comments to the Author**

1. Is the manuscript technically sound, and do the data support the conclusions?

Reviewer #1: Yes

Reviewer #2: Yes

2. Has the statistical analysis been performed appropriately and rigorously?

Reviewer #1: No

Reviewer #2: Yes

3. Have the authors made all data underlying the findings in their manuscript fully available?

Reviewer #1: Yes

Reviewer #2: Yes

4. Is the manuscript presented in an intelligible fashion and written in standard English?

Reviewer #1: Yes

Reviewer #2: Yes

Reviewer #1: Review Report for Manuscript PONE-D-24-55548

Title: Patient Satisfaction in Outdoor Department of Primary Health Care Facilities in Rohingya Refugee Camps in Bangladesh: A Cross-Sectional Study

The manuscript reports on a study evaluating patient satisfaction in the primary healthcare facilities serving the Rohingya refugee camps in Bangladesh. The study employed a cross-sectional design, sampling 810 patients across five health centers with 723 completed responses analyzed. It leveraged the Patient Satisfaction Questionnaire Short Form (PSQ-18) to measure satisfaction across multiple domains. The study addresses an important and underexplored area of healthcare delivery in a crisis setting, providing current data that can inform ongoing health service provision in refugee camps. The use of a structured and validated tool (PSQ-18) to assess patient satisfaction and the application of robust statistical methods are major strengths, contributing to the reliability of the findings. The relatively large sample size and high response rate enhance the study's generalizability to the Rohingya refugee population in Bangladesh.

The abstract has been written nicely and right to the point. The introduction part contains a review of the literature on the issue. The results section is detailed and provides a clear insight into patient satisfaction in the outdoor department of primary health care facilities in Rohingya refugee camps. It successfully highlights various factors influencing patient satisfaction, such as socio-demographic factors, type of visit, and illness. The tables are informative and support the text effectively.

However, following my review, I would like to suggest that the paper needs some revisions, and I think some points need to be addressed before further publication processing. The review comments are given below:

1. In the abstract, specific statistical analysis, such as the “Kruskal–Wallis test” can be mentioned instead of the general term “nonparametric test.”

2. In the introduction section, the authors tried to review some related literature that lacks some basic background information. The authors could consider discussing the major factors contributing to patients’ satisfaction/dissatisfaction with medical consultations and services. Additionally, the current status of health status in Rohingya refugee camps and primary health care services and facilities should be addressed in the introduction section which will indicate the significance of the current study.

3. On page 3 (lines: 60-61) the authors mentioned “A prior study in Nigeria found considerable relationship between a short waiting time and expectations fulfilled by patients during clinic visits [5]”, but the present study didn’t examine the influence of waiting time on patients’ satisfaction instead it is used as one of the scales of the dependent variables. I suggest omitting this sentence and citing a recent study that focuses on various factors influencing patients' satisfaction in Bangladesh (Zakaria, M., Mazumder, S., Faisal, H. M., Zannat, R., Haque, M. R., Afrin, T., ... & Xu, J. (2024). Physician communication behaviors on patient satisfaction in primary care medical settings in Bangladesh. Journal of Primary Care & Community Health, 15, https://doi.org/10.1177/21501319241277.)

4. The major part of the introduction section is written as: A prior study found… Similar research in Iraq revealed… A study on maternal and child health services in rural areas of India revealed… Researchers found… it was found that…which indicates poor English writing. The authors can revise the introduction part to improve the language quality.

5. In the method section, it is necessary to explain more how the authors utilized simple random sampling in selecting study participants. Based on the description provided, it seems more likely that the study employed a form of systematic sampling rather than pure simple random sampling during the second stage of selecting study participants.

6. This section should mention the validity of the scale, such as the translation process of the Patient Satisfaction Questionnaire (PSQ-18) from English to Bangla.

7. On page 6 (lines: 136-138) the author mentioned: “For all question items of PSQ-18, responses were structured across five types; ’Strongly Agree’, ‘Agree’, ‘Uncertain’, ‘Disagree’ and ‘Strongly Disagree’ with values ranging from 1 to 5 (See Table B in S1 File).” According to Table B in S1 File, it should state that for negative statements the scale value was reversed.

8. Existing literature suggests that the Rohingya people are often reluctant to talk with unknown people. How did the researchers handle the issue during data collection? What is the role the Majhi played in this regard?

9. Please rephrase the variable “type of illness” to “perception of the type of illness” since the study did not use medical reports to verify whether conditions were acute or chronic.

10. As the study participants are living in a humanitarian condition and refugee settings, it is expected that there is no significant variability in some of their characteristics. It is also difficult to change their socio-economic status in this current context. Accordingly, the authors could include some interventional variables such as health awareness, NGOs/INGOs intervention, or access to health information. I think it could be mentioned as one of the study’s limitations.

11. The study should perform a multivariate analysis like regression to identify significant predictors since bivariate analysis alone may not sufficiently report influential factors.

12. In the abstract and results section, correct the notation from p = <.001 to p < .001.

13. The surprisingly high patient satisfaction levels reported in this study, despite the ongoing struggles of the Rohingya with healthcare, might be explained by their extreme deprivation in their home country concerning basic health needs.

14. The discussion section should include a paragraph about the implications of the study findings.

Nonetheless, I put some suggestions, still, it is a good piece of erudite work and provides some insights into the patients’ satisfaction in the outdoor department of primary health care facilities in Rohingya refugee camps in Bangladesh that collected data from a large sample. Accordingly, I recommend this article be published after revisions.

Reviewer #2: In the method section it is mentioned that five primary health care facilities were selected randomly as study centers situated across five different Rohingya camps of Bangladesh. However, it is not clear whether the facilities were purely government or government, and NGO supported or purely NGO supported. If the PHC facilities are supported by NGO who were the donor. This information is important to understand the quality of providers, facility readiness, and the monitoring and supportive supervision done by the donor or government etc. as these issues are related to the quality of care provided.

Adding this information regarding PHC facilities will help to understand the quality of care at PHC facilities and will help to set some standards in the recommendations about interpersonal manners, accessibility and convenience. Though authors mentioned in the limitation that provision of emergency allowance/incentives for working in humanitarian settings, working environment inside Rohingya camps, availability of essential diagnostics and medicines for beneficiaries were not evaluated.

**Do you want your identity to be public for this peer review?** For information about this choice, including consent withdrawal, please see our Privacy Policy

Reviewer #1: **Yes: ** Muhammad Zakaria

Reviewer #2: **Yes: ** Dr Khaleda Islam, Director PHC (Retired), DGHS, Bangladesh

---

## [Author Response · Author response to Decision Letter 1]

20 Apr 2025

Response to the Reviewer

Manuscript title: ‘Patient Satisfaction in Outdoor Department of Primary Health Care Facilities in Rohingya Refugee Camps in Bangladesh: A Cross-Sectional Study’

Manuscript reference number: PONE-D-24-55548

We would like to thank the Academic Editor and respected reviewers for their constructive comments and guidance to improve the paper. Following up on the reviewers’ suggestions and recommendations, we have revised the manuscript, and each modification has been highlighted in red. We have modified the sections as the reviewers suggested. Our detailed response as per the clean copy of revised manuscript is found below.

Reviewer Comment:

Author Response:

Reviewer: 1

1. In the abstract, specific statistical analysis, such as the “Kruskal–Wallis test” can be mentioned instead of the general term “nonparametric test.”

Response: Thank you for this helpful suggestion which will enhance statistical precision. Accordingly, we have revised the abstract to explicitly mention the Kruskal–Wallis test instead of the general term “nonparametric test” in Page 2, Line 41

2. In the introduction section, the authors tried to review some related literature that lacks some basic background information. The authors could consider discussing the major factors contributing to patients’ satisfaction/dissatisfaction with medical consultations and services. Additionally, the current status of health status in Rohingya refugee camps and primary health care services and facilities should be addressed in the introduction section which will indicate the significance of the current study.

Response: Thank you for this constructive suggestion. In response, we have elaborated the introduction section in Page 3, Lines 59-62, highlighting the role of interpersonal care capacity and patient-centric behavioral skills of physicians, which have been shown to influence satisfaction more than social, economic and engagement factors. Relevant references have been added accordingly. In addition, as suggested, we reflected the current health status in Rohingya refugee camp in Page 4, lines 92-95 based on recent Public Health Needs Assessment by Health Sector mentioning some key challenges - all of which can influence patient experience.

3. On page 3 (lines: 60-61) the authors mentioned “A prior study in Nigeria found considerable relationship between a short waiting time and expectations fulfilled by patients during clinic visits [5]”, but the present study didn’t examine the influence of waiting time on patients’ satisfaction instead it is used as one of the scales of the dependent variables. I suggest omitting this sentence and citing a recent study that focuses on various factors influencing patients' satisfaction in Bangladesh (Zakaria, M., Mazumder, S., Faisal, H. M., Zannat, R., Haque, M. R., Afrin, T., ... & Xu, J. (2024). Physician communication behaviors on patient satisfaction in primary care medical settings in Bangladesh. Journal of Primary Care & Community Health, 15, https://doi.org/10.1177/21501319241277.)

Response: Thank you for this helpful suggestion. As guided, we have removed the sentence referencing the Nigerian study. Instead, in Page 4, Lines 79-81, we have cited the recommended study by Zakaria et al. (2024) into the section of the introduction where we discuss Bangladeshi literature related to patient satisfaction. This addition strengthens the Bangladesh-specific context of the literature review and supports the rationale for our study in primary health care.

4. The major part of the introduction section is written as: A prior study found… Similar research in Iraq revealed… A study on maternal and child health services in rural areas of India revealed… Researchers found… it was found that…which indicates poor English writing. The authors can revise the introduction part to improve the language quality.

Response: Thank you so much for pointing this out. We acknowledge that the earlier wording in the introduction section became repetitive and informal, damaging language quality. In response, we have thoroughly revised the relevant portion of the introduction to remove redundant phrases such as “a prior study found” and “it was found that” have been replaced with words focusing on the actual findings. The changes are reflected in Page 3-4, Lines 66-79.

5. In the method section, it is necessary to explain more how the authors utilized simple random sampling in selecting study participants. Based on the description provided, it seems more likely that the study employed a form of systematic sampling rather than pure simple random sampling during the second stage of selecting study participants.

Response: Thank you for the insightful comment. We acknowledge the information gap regarding our method of selecting patients as study participants. In our study, we employed systematic random sampling technique in this regard. A single-digit number (3) was randomly selected to determine the starting point, after which every third patient was enrolled. This explanation has been incorporated into the Methodology section (Page 5, Lines 107-111) as well in the Abstract (Page 2, Line 36).

6. This section should mention the validity of the scale, such as the translation process of the Patient Satisfaction Questionnaire (PSQ-18) from English to Bangla.

Response: Thank you for the valuable suggestion. We have clarified the translation process in the revised manuscript in Page 6, Lines 138-141. Linguistic experts within our research team, with prior experience working with the PSQ-18 questionnaire, were involved in the Bengali translation of the tool. Finally, this was finalized through team consensus for ensuring contextual consistency.

7. On page 6 (lines: 136-138) the author mentioned: “For all question items of PSQ-18, responses were structured across five types; ’Strongly Agree’, ‘Agree’, ‘Uncertain’, ‘Disagree’ and ‘Strongly Disagree’ with values ranging from 1 to 5 (See Table B in S1 File).” According to Table B in S1 File, it should state that for negative statements the scale value was reversed.

Response: Thank you for your helpful observation. As suggested, we have updated the manuscript to clarify that for question items that were negatively worded, the scale values were reversed following the scoring guidelines of the PSQ-18 in Page 7, Line 156-57.

This revision has been made in the Methods section, and Table B in the S1 File already reflects the correct scoring structure.

8. Existing literature suggests that the Rohingya people are often reluctant to talk with unknown people. How did the researchers handle the issue during data collection? What is the role the Majhi played in this regard?

Response: Thank you for this important observation. We have addressed these issues in Page 6-7, Lines 142-147. We recruited data collectors with substantial prior experience working in the Rohingya camps and fluency in the Chattogram dialect, closely resembling the Rohingya language. In addition, community leaders (Majhi) related with the study sites were informed about the study’s background, objectives, data collection procedures, voluntary nature of participation, confidentiality of responses and the expected outcomes, which helped build trust and support within the community.

9. Please rephrase the variable “type of illness” to “perception of the type of illness” since the study did not use medical reports to verify whether conditions were acute or chronic.

Response: Thank you for this insightful suggestion. We agree with your observation and have rephrased the variable from “type of illness” to “perception of the type of illness” in the narrative and tables generated as this classification was based on patient self-report, not clinical verification.

10. As the study participants are living in a humanitarian condition and refugee settings, it is expected that there is no significant variability in some of their characteristics. It is also difficult to change their socio-economic status in this current context. Accordingly, the authors could include some interventional variables such as health awareness, NGOs/INGOs intervention, or access to health information. I think it could be mentioned as one of the study’s limitations.

Response: Thank you for this valuable suggestion and we absolutely agree with you. We incorporated your advice as one of our limitations in Page 20, Lines 352-355. However, these variables were beyond the scope of our current study.

11. The study should perform a multivariate analysis like regression to identify significant predictors since bivariate analysis alone may not sufficiently report influential factors.

Response: Thank you for your thoughtful recommendation. Accordingly, we have performed multivariate linear regression analysis to identify independent predictors of patient satisfaction across its various dimensions. The results have been presented in Page 16, Table 5. We have also mentioned in Abstract section (Page 2, Line 41-42), elaborated in the Results section (Page 16, Lines 265-270) as well as in Discussion section (Page 18-19, Lines 317-333).

12. In the abstract and results section, correct the notation from p = <.001 to p < .001.

Response: Thank you for pointing this out. We have corrected the notation in both the abstract and results section in accordance with standard statistical reporting conventions.

13. The surprisingly high patient satisfaction levels reported in this study, despite the ongoing struggles of the Rohingya with healthcare, might be explained by their extreme deprivation in their home country concerning basic health needs.

Response: Thank you for the insightful comment. We have addressed the surprisingly high satisfaction levels in the Discussion section (Page 17, Lines 275-282). This may be explained by the contrast with past deprivation in Myanmar, as well as the structured service delivery and strong coordination among humanitarian actors in the camps.

14. The discussion section should include a paragraph about the implications of the study findings.

Response: Thank you for this thoughtful suggestion. We have now added a dedicated paragraph in the Discussion section (Page 20, Lines 356-362) on practical implications of our findings. This paragraph highlights how key domains of patient satisfaction—such as interpersonal manner, communication, and time spent with doctors—can inform policy development, guide targeted interventions, and support effective resource allocation in humanitarian health settings.

Reviewer 2:

Reviewer Comment:

In the method section it is mentioned that five primary health care facilities were selected randomly as study centers situated across five different Rohingya camps of Bangladesh. However, it is not clear whether the facilities were purely government or government, and NGO supported or purely NGO supported. If the PHC facilities are supported by NGO who were the donor. This information is important to understand the quality of providers, facility readiness, and the monitoring and supportive supervision done by the donor or government etc. as these issues are related to the quality of care provided. Adding this information regarding PHC facilities will help to understand the quality of care at PHC facilities and will help to set some standards in the recommendations about interpersonal manners, accessibility and convenience. Though authors mentioned in the limitation that provision of emergency allowance/incentives for working in humanitarian settings, working environment inside Rohingya camps, availability of essential diagnostics and medicines for beneficiaries were not evaluated.

Response: Thank you for this thoughtful and important observation. We agree that clarifying the nature of the selected primary health care facilities adds critical context regarding the quality of service provision and helps interpret the satisfaction levels observed. In response, we have revised the manuscript in Methodology section (Page 5, Lines 105-106). The selected health facilities were operationalized by non-government organization with financial support from international and UN agencies. In addition, we have reflected this observation in the Discussion section (Page 17, Lines 278-282), noting that the structured nature of the service delivery model—supported by national NGOs and humanitarian partners—may have contributed to the high levels of satisfaction reported in our study.

Finally, We strongly believe that the reviewers’ comments have helped us to improve the presentation, readability and technicalities of the manuscript. We thank you again for your valuable comments.

Thank you.

---

## [Decision Letter · Decision Letter 1]

30 Oct 2025

Patient satisfaction in outdoor department of primary health care facilities in Rohingya refugee camps in Bangladesh: a cross-sectional study

PONE-D-24-55548R1

Dear Dr. Hawlader,

We’re pleased to inform you that your manuscript has been judged scientifically suitable for publication and will be formally accepted for publication once it meets all outstanding technical requirements.

Kind regards,

Sk Md Mamunur Rahman Malik

Academic Editor

PLOS ONE

Additional Editor Comments (optional):

The article can be published. However, I have some observations, which I have included in the decision letter. Please go through those observations and revise your manuscript during the editing process.

1. The "method" section needs to be restructured. Please start the section with how you defined the single proportion sample size and then describe how the respondents were selected to reach the desired number of your sample. I am not sure why a confidence interval of 97% was selected instead of 95% or if you wanted more precision, you could have used a CI of 99%. An explanation needs to be provided for using a 97% CI which is not a standard practice. I also feel that the formula that you have used for calculating the sample size is a bit different than what the usual practice is. However, please provide a description of what Z, p, q, and d stand for in your formula. Either you show the calculation in the text of how you reached the magic number of 723 using this formula or simply leave it but explain the meaning of all parameters you have used to calculate the sample size.

2. Please include the non-response rate in the section on "data collection" putting the number of refusals.  

3. The statistical values (mean, SD, and p-value) you have included in tables (1/4) are not in the right sequence. The "p-value" should always be presented after the "mean value" for readers to understand the statistical significance of the calculated mean values for the variables.  I am hoping that the editorial team of the Journal will assist you in reformatting the tables with correctly showing how the statistical values need to be presented. 

4. Please check line 347 and lines 348 to 352 of the "Discussion" section. The language used in line 347 is not understandable. For lines 348 to 352, my observation is that while availability of diagnostic and other services in health facilities can contribute to patient satisfaction, I am not convinced that perception of healthcare workers, their incentives, and other work-related conditions of HCWs have any role in the satisfaction of patients unless you want to explain your rationale.

5. I haven't found any narration or text regarding "quality of care" and how it relates to satisfaction of patients in healthcare settings. A general comment would be useful. 

6. I think the authors have presumed that every reader will know the "Rohingya refugees" and their healthcare situation. A brief description of who these refugees are, how big the size of this refugee problem is, and why it is important to know the patient satisfaction of these communities while understanding the quality of care they receive is an important attribute that I did not find in the manuscript.

6. Finally, it is intriguing that the majority of care seekers were females. Any reason why this is? Where are the males? Do they not seek health care during the time data were collected, or is there any anthropological or demographic reason for this?

7. In the "Discussion" section, it will be good to know the author's perspective on why some variables were not associated with higher satisfaction compared to 2/3 variables that were positively associated. 

Thank you and I congratulate the  authors for getting their paper accepted. 

Reviewers' comments:

Reviewer's Responses to Questions

**Comments to the Author**

Reviewer #1: All comments have been addressed

Reviewer #2: All comments have been addressed

2. Is the manuscript technically sound, and do the data support the conclusions?

Reviewer #1: Yes

Reviewer #2: Yes

3. Has the statistical analysis been performed appropriately and rigorously?

Reviewer #1: Yes

Reviewer #2: Yes

4. Have the authors made all data underlying the findings in their manuscript fully available?

Reviewer #1: Yes

Reviewer #2: Yes

5. Is the manuscript presented in an intelligible fashion and written in standard English?

Reviewer #1: Yes

Reviewer #2: Yes

Reviewer #1: I appreciate the authors’ thoughtful and thorough responses to the reviewer comments. The revisions have adequately addressed all the concerns raised, and the manuscript has been significantly improved in clarity, rigor, and relevance. I recommend acceptance of the paper in its current form.

Reviewer #2: All the review comments were addressed first in the Methodology (study settings and subjects) section and secondly in the discussion section.

**Do you want your identity to be public for this peer review?** For information about this choice, including consent withdrawal, please see our Privacy Policy

Reviewer #1: **Yes: ** Muhammad Zakaria

Reviewer #2: **Yes: ** Khaleda Islam

---

## [Editor Report · Acceptance letter]

PONE-D-24-55548R1

PLOS One

Dear Dr. Hawlader,

I'm pleased to inform you that your manuscript has been deemed suitable for publication in PLOS One. Congratulations! Your manuscript is now being handed over to our production team.

Kind regards,

on behalf of

Dr. Sk Md Mamunur Rahman Malik

Academic Editor

PLOS One